# Alpha-Keratin, Keratin-Associated Proteins and Transglutaminase 1 Are Present in the Ortho- and Parakeratinized Epithelium of the Avian Tongue

**DOI:** 10.3390/cells11121899

**Published:** 2022-06-11

**Authors:** Kinga Skieresz-Szewczyk, Hanna Jackowiak, Marek Skrzypski

**Affiliations:** 1Department of Histology and Embryology, Faculty of Veterinary Medicine and Animal Science, Poznan University of Life Sciences, Wojska Polskiego 71C, 60-625 Poznan, Poland; hanna.jackowiak@up.poznan.pl; 2Department of Animal Physiology, Biochemistry and Biostructure, Faculty of Veterinary Medicine and Animal Science, Poznan University of Life Sciences, Wołyńska 35, 60-637 Poznan, Poland; marek.skrzypski@up.poznan.pl

**Keywords:** cornification, parakeratinized epithelium, orthokeratinized epithelium, alpha-keratin, KAPs, tongue, birds

## Abstract

The lingual mucosa in birds is covered with two specific types of multilayered epithelia, i.e., the para- and orthokeratinized epithelium, that differ structurally and functionally. Comprehensive information on proteins synthesized in keratinocyte during their cytodifferentiation in subsequent layers of multilayered epithelia in birds concerns only the epidermis and are missing the epithelia of the lingual mucosa. The aim of the present study was to perform an immunohistochemical (IHC) and molecular analysis (WB) of bird-specific alpha-keratin, keratin-associated proteins (KAPs), namely filaggrin and loricrin, as well as transglutaminase 1 in the para- and orthokeratinized epithelium covering the tongue in the domestic duck, goose, and turkey. The results reveal the presence of alpha-keratin and KAPs in both epithelia, which is a sign of the cornification process. In contrast to the epidermis, the main KAPs involved in the cornification process of the lingual epithelia in birds is loricrin. Stronger expression with KAPs and transglutaminase 1 in the orthokeratinized epithelium than in the parakeratinized epithelium may determine the formation of a more efficient protective mechanical barrier. The presence of alpha-keratin, KAPs, and transglutaminase 1 epitopes characteristic of epidermal cornification in both types of the lingual epithelia may prove that they are of ectodermal origin.

## 1. Introduction

Keratinocytes forming the multi-layered epithelium undergo specific cytodifferentiation leading to the formation of a protective superficial layer. So far, the names keratinization and cornification in publications have been used interchangeably. Nowadays, research emphasizes that these are two different processes. In general, during the keratinization process, bundles of the intermediate filament build of the cytokeratins are accumulated in the cytoplasm of keratinocytes, whereas during cornification, besides intermediate filament keratins, other proteins such as keratin-associated proteins (KAPs) are also produced [1,2,3].

Studies on the cornification of vertebrate epithelium most often describe the integument epithelium and their cornified structures. In mammals, there are generally two types of cornification, i.e., soft cornification, which affects the epidermis, and hard cornification, which occurs in the hair, nails, and claws [4,5].

During soft cornification, intermediate cytokeratin filaments composed generally of CK1 and CK10 are collected in the cytoplasm of the keratinocytes of the spinous, granular, and cornified layers of the epidermis [6]. The synthesis of keratin-associated proteins begins as early as in the spinous layer, where filaggrin and loricrin are formed. Both proteins form specific keratohyalin granules in the form of irregular F-granules and smaller L-granules in the granular layer of the epidermis [6]. Filaggrin is involved in the assembly of cytokeratin filaments into thick bundles that completely fill the cytoplasm of the cells of the cornified layer of the epithelium [7,8,9]. On the other hand, loricrin along with other keratin-associated proteins, such as involucrin, envoplakin, and periplakin, are arranged by transglutaminase 1 (TGM-1) along the inner surface of the cell membrane and form the protein part of the so-called cornified cell envelope [10,11,12]. During the differentiation of subsequent layers of keratinocytes, lipids such as free fatty acids, cholesterol, and its esters, and different glucosylceramide are also synthesized in the cytoplasm of the granular layer cells and are accumulated in lamellar bodies [13]. In the higher parts of the granular layer of the epidermis, lamellar bodies move under the cell membrane, where they fuse with the cell membrane, and lipids are extruded outside the keratinocytes, where they form the lipid part of the cornified cell envelope. The task of the cornified layer of the epidermis constructed in this way is mechanical protection and the creation of a permeability barrier [13,14].

The process of cornification of multilayered epithelia of birds takes place in a different way. It is important to note that, in the cytoplasm of the keratinocytes of the apteric and scale epidermis, bird-specific cytokeratins called alpha- and beta-keratins are synthesized [15,16,17]. Next to alpha- and beta-keratin, the presence of filaggrin and loricrin was revealed [15,16]. It should be noted that these proteins do not form keratohyalin granules [15]. The research of Alibardi [15] and Alibardi and Toni [16] also indicated the presence of TGM-1, which occurs only in the transitional layer in the apteric epidermis and is absent in the scale epidermis.

The cornification process takes place not only in the epidermis and its cornified structure but also in the multilayered epithelium of the oral mucosa. In mammals, the process of cornification in the oral cavity is most often described with reference to the strongly cornified filiform papillae. On the anterior and posterior surfaces of the papillae, the phenomenon of the dual pattern of cornification is observed, i.e., soft cornification and hard cornification [18,19].

Studies on the cornification process carried out so far in the avian beak cavity, namely on the tongue, indicate the presence of two types of cornified epithelia, i.e., para- and orthokeratinized epithelium [20,21,22,23]. It should be emphasized that these epithelia are composed of the basal layer, the intermediate layer, and the cornified layer [20,21,22,23]. However, as in the epidermis of birds, keratohyalin granules have never been observed. The parakeratinized epithelium is characterized by a greater height compared to the orthokeratinized epithelium, and the keratinocytes in the cornified layer have still cell nuclei [21,22,23]. The orthokeratinized epithelium is lower with a relatively thick cornified layer, in which the keratinocytes lack cell nuclei [21,22,23]. It is important to remember that both types of epithelia functionally cover different areas of the mucosa of the bird’s tongue. Generally, the parakeratinized epithelium occurs in places involved in the transport of food to the esophagus, i.e., the dorsal surface of the tongue, whereas the orthokeratinized epithelium is present in places directly involved in food intake, namely the ventral surface of the lingual apex with the plate of the lingual nail and conical papillae present on the edges of the tongue in Anseriformes or conical papillae present on the border of the lingual body and root [21,22,23].

Analyses of the components synthesized during cornification of the para- and orthokeratinized epithelium of the tongue in birds concern only the presence of alpha- and beta-keratin, which are accumulated in the cytoplasm of keratinocytes, similar to the apteric and scale epidermis [24,25,26].

So far, there are no data about other components, i.e., keratin-associated proteins and TGM-1.

In the current study, the following research hypothesis was assumed: The same proteins are involved in the cornification process of the ortho- and parakeratinized epithelium of the tongue in birds as in the cornification of the epidermis and its appendages.

The aim of the present study is to perform, for the first time, an immunohistochemical (IHC) and molecular analysis of alpha-keratin and keratin-associated proteins, namely filaggrin and loricrin, as well as TGM-1, in the para- and orthokeratinized epithelium of the lingual mucosa in the domestic duck, goose, and turkey. These avian species are characterized by a different microstructure of the lingual mucosa and diverse food strategies regarding consumed food (plant food immersed in water, green parts of terrestrial plants, grains), mechanisms of food intake (filter-feeding, grazing, pecking), and food transport to the esophagus (under-tongue transport, over tongue transport, catch-and-throw transport) [27,28,29,30].

The obtained results will be also used in functional comparative analyses of the cornification of the epithelium of ectodermal origin, namely the epithelium of the mucosa of the bird’s beak cavity and the epidermis and scales.

## 2. Materials and Methods

The research material was tongues of adult domestic duck, domestic goose, and domestic turkey collected after slaughter from local breeders. According to Polish law and EU directive no 2010/63/EU, the present study does not require approval of the Local Ethical Committee for Experiments on Animals in Poznan.

Three tongues were collected for each species to proceed with morphological and immunohistochemical analysis. The tongues were rinsed in saline and fixed in 4% buffered formalin. From the tongues, the orthokeratinized epithelium was collected from the ventral surface of the lingual apex and the parakeratinized epithelium from the dorsal surface of the lingual body. Fixed tissue samples were subjected to the standard tissue preparation procedure, i.e., dehydrated in a series of increasing concentrations of ethanol (70–96%) and routinely embedded in Paraplast^®^ (Sigma-Aldrich, Darmstadt, Germany). Paraplast blocks were cut into serial sections of 4.5–5 μm. Tissue sections were stained using the Masson–Goldner trichrome histological staining technique [31]. Tissue samples for immunohistochemistry staining proceeded according to the protocol by Skieresz-Szewczyk et al. [25] in which they were deparaffinized for 12 h at 60 °C and washed in three changes of xylene and a series of decreasing alcohol concentrations (99.8–80%). To expose epitopes, histological sections were placed in the buffer DakoTarget Retrieval Solution, pH 5.9 (Dako, Santa Clara, CA, USA) in a water bath type WB-4 MS (Biosan, Jozefów, Poland) at 97 °C for 40 min. Then samples were cooled for 20 min at room temperature and washed in double-distilled water. Nonspecific antibody binding was blocked with a 3% hydrogen peroxide solution at room temperature for 10 min. After that, the tissue samples were washed in two changes of double-distilled water and phosphate-buffered saline (PBS, pH 7.4) for 10 min. Tissue samples were incubated with the primary antibody at 37 °C in a humid chamber for 30 min. The following antibodies were used: (1) Cytokeratin clone AE1/3 (dilution 1:50, cat. M3515, Dako, Santa Clara, CA, USA), (2) anti-filaggrin (dilution 1:50, cat. ab17808, Abcam, Cambridge, UK), (3) anti-loricrin (dilution 1:500, cat. ab85679, Abcam, Cambridge, UK), and (4) anti-TGM1 (dilution 1:100, cat. Ab183351, Abcam, Cambridge, UK). The antibodies were diluted with Dako Antibody Diluent (Dako, Santa Clara, CA, USA). After incubation, histological sections were rinsed in PBS (pH 7.4) for 10 min and incubated in a humid chamber for 20 min at 37 °C with goat anti-rabbit/mouse IgG (Polymer Labeled HRP-Anti-Rabbit/Mouse; Dako, Santa Clara, CA, USA). Afterward, tissue samples were washed in PBS (pH 7.4) for 10 min. The DAB solution (Dako, Santa Clara, CA, USA) was used to visualize the binding of the secondary antibodies. The samples were stained in Mayer’s hematoxylin for 5 min and dehydrated in a series of increasing alcohol concentrations (80–99.8%) and acetone. For negative controls, the first antibody was omitted, and instead, the tissue samples were incubated with PBS (pH 7.4).

Observations of the histological sections were performed using an Axioscope2plus light microscope (Zeiss, Göttingen, Germany). Five documented histological sections from Masson–Goldner staining were used to proceed with the morphometric analysis. The Multiscan computer morphometric system (ver. 10.2, CSS, Warsaw, Poland) was used to proceed with four measurements on each histological section, which gave a total of 20 measurements to estimate the mean value of the height of the epithelium, its cornified layer, and the height and width of the connective tissue papillae.

For the Western blot analysis, three tongues for each species were used to collect the samples of the parakeratinized epithelium and orthokeratinized epithelium. Epithelial fragments were homogenized in RIPA buffer (50 mmol/L Tris–HCl, pH 8.0 with 150 mmol NaCl, 1.0% NP-40, 0.5% sodium deoxycholate, 0.1% SDS), containing the cOmplete™ Protease Inhibitor Cocktail (Roche Diagnostics, Penzberg, Germany). Homogenates epithelia were centrifuged at 14,000× *g* for 10 min., and the protein concentration was determined using the Pierce BCA Protein Assay Kit (Thermo Scientific, Wilmington, DE, USA). Protein samples (20 μg per lane) were separated on 5–12% Tris-HCl SDS-PAGE gel and blotted onto a nitrocellulose membrane (BioRad, Hercules, CA, USA). Nitrocellulose membranes were blocked for 1 h at room temperature in 5% bovine serum albumin in TBST buffer. Then, membranes were incubated with primary antibodies for 24 h at 4 °C (1:100 dilution, Anti-loricrin cat. SAB2108557, Sigma-Aldrich, Darmstadt, Germany; Anti-TGM1 cat. SAB1400277, Sigma-Aldrich, Darmstadt, Germany; Anti-cytokeratin clone AE1/3 cat. MAB3412, Sigma-Aldrich, Darmstadt, Germany; Anti-filaggrin cat. ab81468, Abcam, Cambridge, UK). After that, the nitrocellulose membrane was washed 3 times using TBST buffer and incubated with a secondary antibody, diluted at 1:5000 for 1 h at room temperature, and washed again 3 times using TBST buffer. Signals were visualized using the Amersham ECL prime Western Blotting Detection Reagent (GE Healthcare Life Sciences, Little Chalfont, UK) on the VersaDoc Imaging System (BioRad Laboratories, Munich, Germany).

## 3. Results

### 3.1. Morphological Observation

The microscopic observation of the histological section of the orthokeratinized epithelium on the ventral surface of the lingual apex and the parakeratinized epithelium on the dorsal surface of the lingual body in the duck, goose, and turkey reveals that they are built on the basal layer, intermediate layer, and cornified layer (Figure 1a–c and Figure 2a–c). The average height of the orthokeratinized epithelium is 241.8 µm in the duck, 339.9 µm in the goose, and 175.9 µm in the turkey. The height of the cornified layer is approximately 70.9 µm, 103.8 µm, and 70.4 µm, respectively. In the case of the parakeratinized epithelium, the average height is 540.9 µm in the duck, 520 µm in the goose, and 1114.3 µm in the turkey, and the cornified layer is 89.2 µm, 64.2 µm, and 118.5 µm in height, respectively.

The lamina propria of the lingual mucosa forms connective tissue papillae in both types of the epithelia, which vary in density, length, and arrangement within the epithelium (Figure 1a–c and Figure 2a–c). In the orthokeratinized epithelium, the connective tissue papillae are thin ca. 15.8 µm in duck, 17.8 µm in goose, and 17.9 µm in turkey. They are arranged perpendicular to the basal membrane of the epithelium and reach the level of the lower part of the intermediate layer (Figure 1a–c). The mean height of the connective tissue papillae in the orthokeratinized epithelium is 40.5 µm in duck, 68.1 µm in goose, and 14.1 µm in turkey. In the parakeratinized epithelium, the thick connective tissue papillae in the duck and goose are also located perpendicular to the basal membrane and reach the level of the upper part of the intermediate layer (Figure 2a,b). Their average height is 199.5 µm and 199.8 µm, respectively, while the width is ca. 31.6 µm in duck and 33.5 µm in goose. On the other hand, in turkey, in the parakeratinized epithelium, the connective tissue papillae 715.7 µm in height are arranged obliquely and reach the cornified layer of the epithelium (Figure 2c). The width of the connective tissue papillae in turkey is higher than in duck and goose and reaches ca. 52.4 µm.

The cells in the basal layer of both types of cornified epithelia are elongated with oval nuclei and 1–3 nucleoli and the cytoplasm of these cells stains pink according to the Masson–Goldner stain (Figure 1a–c and Figure 2a–c).

The intermediate layer in the ortho- and parakeratinized epithelium is divided into two parts due to the shapes of cells and their nuclei. The lower part of the intermediate layer of both types of epithelia is composed of polygonal cells with round or oval cell nuclei and 1–2 nucleoli (Figure 1a–c and Figure 2a–c). In goose, cell nuclei almost completely fill the cell cytoplasm. The cells of the lower part of the intermediate layer of the orthokeratinized epithelium in all studied species of birds and the parakeratinized epithelium in duck and goose, due to the perpendicular arrangement of connective tissue papillae, are arranged horizontally. In contrast, in the parakeratinized epithelium of turkey, due to obliquely arranged connective tissue papillae, both the basal and lower part of the intermediate layer is characterized by a wavy arrangement, which disturbs the typical horizontal localization of the cell layers, and on the cross-section between the connective tissue papillae, cells of the basal and lower part of the intermediate layer are present (Figure 2c). Cells in the upper part of the intermediate layer are flattened and have elliptical or flattened nuclei with 1–2 nucleoli (Figure 1a–c and Figure 2a–c). In the orthokeratinized epithelium of the three studied species of birds and in the parakeratinized epithelium of duck and goose, the cells of the upper part of the intermediate layer are arranged horizontally, while in the turkey, they are arranged horizontally or obliquely. The cytoplasm of the cells of both parts of the intermediate layer in the parakeratinized epithelium stains uniformly pink (Figure 2a–c). Differences in the color of the cytoplasm within two parts of the intermediate layer were observed in the orthokeratinized epithelium of the studied bird species. The cytoplasm of the cells of the lower part of the intermediate layer stains pink and the upper part of the intermediate layer is dark pink (Figure 1a–c).

The cornified layer of the ortho- and parakeratinized epithelium in the studied bird species is composed of flattened cells. In the orthokeratinized epithelium, cells of the cornified layer are devoid of cell nuclei, and their cytoplasm stains red as a result of Masson–Goldner staining (Figure 1a–c). The superficial cells of the cornified layer of the orthokeratinized epithelium exfoliate in the form of individual scales (Figure 1a–c).

In the parakeratinized epithelium, cells of the cornified layer have nuclei with condensed chromatin (Figure 2d–f). The microscopic observations of the parakeratinized epithelium stained with the Masson–Goldner method revealed differences in the color of the cytoplasm of the cornified layer in the studied bird species (Figure 2d–f). The cytoplasm of 3–4 superficial cells of the cornified layer in duck is stained pink, and the cytoplasm of the underlying cells is stained red (Figure 2d). In goose, 1–2 layers of the superficial cells of the cornified layer are stained light pink, while the cytoplasm of the lower cells is stained red (Figure 2e). The cornified layer in turkey is composed of 5–6 layers of superficial cells with a light pink cytoplasm that exfoliate extensively (Figure 2f). The cells of the cornified layer, which are situated at the lower level and make up approximately 10–12 layers, are characterized by a different color of the cytoplasm. The cell cytoplasm is stained alternately in red or light pink (Figure 2f).

### 3.2. Immunohistochemical Analysis

Results of the immunohistochemical staining with AE1/3, filaggrin, loricrin, and TGM-1 antibodies are presented in Table 1, Table 2, Table 3 and Table 4.

#### 3.2.1. Alpha-Keratin Analysis

IHC staining with AE1/3 antibody, which detects alpha-keratin in the multi-layered epithelia in birds revealed a positive, diffuse staining reaction, in the whole-cell cytoplasm in all epithelial layers of the ortho- and parakeratinized epithelium in examined bird species (Figure 3a–c and Figure 4a–c, Table 1).

In the orthokeratinized epithelium, the cytoplasm of the basal, lower, and upper parts of the intermediate layer cells in duck, goose, and turkey shows a strong color reaction with the AE1/3 antigen (Figure 3a–c, Table 1). In the case of the cornified layer of the orthokeratinized epithelium, immunohistochemical staining shows a weak color response in all tested bird species (Figure 3a–c, Table 1).

In the parakeratinized epithelium, the level of color reactions with AE1/3 in all layers of the epithelium in the three bird species studied is generally strong (Figure 4a–c, Table 1), wherein medium staining is observed in the basal layer of the parakeratinized epithelium in turkey (Figure 4c, Table 1). Interesting results of IHC reactions with the AE1/3 antigen are recorded in the cornified layer of the parakeratinized epithelium in duck, goose, and turkey. The superficial cells of the cornified layer, which form 1–2 layers in duck and goose and 5–6 layers of cells in turkeys, are characterized by a strong color reaction, while the cytoplasm of cells located lower in the cornified layer of the parakeratinized epithelium in all studied species of birds is characterized by a medium color reaction (Figure 4d–f, Table 1).

#### 3.2.2. Filaggrin Analysis

Immunohistochemical staining with the anti-filaggrin antigen shows expression at weak or medium levels in the cell cytoplasm of some layers of ortho- and parakeratinized epithelium in duck, goose, and turkey (Figure 5a–c and Figure 6a–c, Table 2).

In general, in the orthokeratinized epithelium of the studied bird species, positive color reactions are observed only in the intermediate layer in the granular pattern (Figure 5a–c, Table 2). The intensity of the color reaction with anti-filaggrin varies between species. In duck and turkey, only the upper part of the intermediate layer shows medium and weak staining, respectively (Figure 5d,f, Table 2). In goose, both parts of the intermediate layer show expression at a medium level (Figure 5e, Table 2). However, observations of the cytoplasm of the upper part of the orthokeratinized epithelium in goose show that cells arranged in 2–3 layers just below the cornified layer have weak or no color reaction (Figure 5e, Table 2).

In the parakeratinized epithelium, a positive staining reaction with the anti-filaggrin is demonstrated not only in the intermediate layer but also in the basal and cornified layers in duck and goose and in the cornified layer in turkey (Figure 6a–c, Table 2). The reaction in the basal and intermediate layers is in a granular pattern in the domestic duck and goose and is diffuse in the domestic turkey.

The cornified layer in all examined birds is characterized by a diffuse reaction. In duck and goose, weak color reactions are recorded in all layers of the parakeratinized epithelium (Figure 7a,b and Figure 8a–c, Table 2).

However, attention should be paid to the varied color staining in the cornified layer of the parakeratinized epithelium in these bird species. In duck, 2–3 layers of superficial cells show a positive expression at the weak level and the lower cells do not show an expression with anti-filaggrin (Figure 7a, Table 2). On the other hand, in goose, 2–3 layers of superficial cells of the cornified layer show a medium color reaction, and the remaining cells of the lower cornified layer show weak staining (Figure 8a, Table 2).

In the case of the turkey, weak color reactions are observed in the entire lower part of the parakeratinized epithelium, which can be seen in the form of bands arranged between the connective tissue papillae (Figure 9b,c, Table 2). In the upper part of the intermediate layer, only weak DAB staining is shown in 2–3 layers of cells located just below the cornified layer (Figure 9a, Table 2). The cornified layer of the parakeratinized epithelium in the turkey shows a weak color response in 3–4 layers of superficial cells (Figure 9a, Table 2). In contrast, the lower cornified layer cells do not show any expression with anti-filaggrin (Figure 9a, Table 2).

#### 3.2.3. Loricrin Analysis

IHC staining with the anti-loricrin antibody shows an expression in the cytoplasm of cells of all layers of the ortho- and parakeratinized epithelium in duck, goose, and turkey (Figure 10a–c and Figure 11a–c, Table 3).

The reactions with anti-loricrin in the orthokeratinized epithelium in all examined bird species are generally diffuse in the whole cytoplasm. In the orthokeratinized epithelium, the basal layer shows a weak color reaction in all the studied bird species (Figure 10d–f, Table 3). A differential color staining with anti-loricrin is observed in the intermediate layer (Table 3). In duck, the lower part of the intermediate layer has a medium color response and the upper part of the intermediate layer shows a strong color response (Figure 10d, Table 3). In contrast, in goose, both parts of the intermediate layer of the orthokeratinized epithelium show a strong expression with anti-loricrin (Figure 10e, Table 3). In turkey, a strong reaction is noted only in the cytoplasm of the cells of the upper part of the intermediate layer, while the cytoplasm of the cells in the lower part of the intermediate layer shows a weak reaction (Figure 10f, Table 3). The cornified layer of the orthokeratinized epithelium as a result of IHC staining with anti-loricrin is characterized by a weak staining reaction in duck, goose, and turkey (Figure 10a–c, Table 3).

In the parakeratinized epithelium in the duck and goose, reactions with ani-loricrin are in a granular pattern in the basal and intermediate layers and diffuse in the cornified layer, while in the turkey, the reactions are generally diffuse in the cell cytoplasm. The cytoplasm of the basal layer cells shows a weak color reaction with anti-loricrin only in the turkey, and in the duck and goose, a medium color response is recorded (Figure 11a–c, Figure 12b, Figure 13c and Figure 14c, Table 3).

The intermediate layer of the parakeratinized epithelium is generally characterized by a medium DAB staining in the duck, goose, and turkey (Figure 11a–c, Figure 12a,b, Figure 13b,c and Figure 14a–c, Table 3). Wherein, in the goose and turkey, attention should be paid to the weak expression in the cytoplasm of cells in the upper part of the intermediate layer, located just below the cornified layer (Figure 13a and Figure 14a, Table 3).

In the case of the cornified layer, a generally weak color reaction with anti-loricrin is recorded in the three tested bird species (Figure 11a–c and Figure 14a, Table 3). However, an interesting observation is seen in the cytoplasm of 2–3 superficial cells of the cornified layer of duck and goose, which shows a stronger color reaction in relation to the lower cells (Figure 12a and Figure 13a, Table 3).

#### 3.2.4. TGM-1 Analysis

The IHC reaction with TGM-1 indicates the expression in all layers of the orthokeratinized epithelium in the three tested bird species (Figure 15a–c, Table 4).

In the case of the parakeratinized epithelium, positive expression in all epithelial layers is recorded in the duck and turkey (Figure 16a–c, Table 4), while in the goose, a positive color reaction is only present in the cornified layer of the parakeratinized epithelium (Figure 16a–c, Table 4). The reaction with TGM-1 in both epithelia and in all examined birds is in a granular pattern.

In general, the basal layer, the lower part of the intermediate layer, and the cornified layer of the orthokeratinized epithelium in the duck, goose, and turkey show weak DAB staining with TGM-1 (Figure 15d–f, Table 4). In contrast, the upper part of the intermediate layer of the orthokeratinized epithelium shows a medium color reaction in all the studied bird species (Figure 15d–f, Table 4).

In the parakeratinized epithelium, the IHC staining differs between species (Figure 16a–c, Table 4). In the duck, weak expression is observed in the basal and intermediate layers and medium expression in the cornified layer (Figure 17a,b, Table 4). In the goose, a medium color reaction is observed only in 2–3 layers of superficial cells of the cornified layer in the parakeratinized epithelium (Figure 17c, Table 4). The lower cells of the cornified layer and the cells of the intermediate and basal layers do not show positive DAB staining with TGM-1 in the goose (Figure 16b and Figure 17c, Table 4).

In contrast, in the turkey, weak color reactions are noted in the cytoplasm of the cells of the intermediate and cornified layers (Figure 18a–c, Table 4). The basal layer of the parakeratinized epithelium in the turkey does not show any expression (Figure 18d, Table 4).

### 3.3. Western Blot Analysis

Immunoblots of the ortho- and parakeratinized epithelium reveal the presence of polypeptide bands at a different molecular weight, which produces a staining reaction with the antibody anti-AE1/3 detecting alpha-keratin, anti-filaggrin, anti-loricrin, and anti-TGM-1 in the three examined birds species (Figure 19, Figure 20, Figure 21 and Figure 22, Table 5).

The polypeptides with molecular weight at 45 kDa strongly reacted with the AE1/3 antiserum in the ortho- and parakeratinized epithelium in the duck, goose, and turkey (Figure 19a,b, Table 5). Additionally, in the parakeratinized epithelium, a strong reaction with the AE1/3 is also detected at 47 kDa in the three studied bird species (Figure 19b, Table 5). The higher molecular weight bands in the range of 57–60 kDa produce a weak or medium reaction with AE1/3 both in the ortho- and parakeratinized epithelium, respectively (Figure 19a,b, Table 5).

Immunoblots incubated with the antibody anti-filaggrin reveal that the polypeptides bands at 30 kDa and 110 kDa in all examined bird species and at 100 kDa in the duck exhibit a strong reaction with the antibody anti-filaggrin in the orthokeratinized epithelium (Figure 20a, Table 5). Interestingly, the strong response to the filaggrin antiserum occurs in the bands at 230 kDa in the turkey (Figure 20a, Table 5). Other bands at 20 kDa exhibit a weak staining reaction only in the duck and goose (Figure 20a, Table 5). In the case of the parakeratinized epithelium, only the polypeptide band at 30 kDa strongly reacts with the filaggrin antiserum in all studied bird species (Figure 20b, Table 5). Other bands at 70 kDa and 110 kDa, with a strong reaction to the filaggrin, are only visible in the duck (Figure 20b, Table 5). Immunoblots also reveal the presence of a weak staining reaction with the antibody anti-filaggrin at 20 kDa in the duck and goose and 220–230 kDa in the goose and turkey (Figure 20b, Table 5).

The polypeptides with a molecular weight of 33–35 kDa strongly react with the loricrin antiserum in both examined epithelia in all studied bird species (Figure 21a,b, Table 5). Other bands in the orthokeratinized epithelium with a molecular weight of approximately 60 kDa and 100 kDa react less intensely with the loricrin antiserum in all bird species (Figure 21a, Table 5). In the case of the parakeratinized epithelium, the weak reaction with loricrin is visible with polypeptide bands at 80 kDa in the duck and 100 kDa in all examined bird species (Figure 21b, Table 5). The strong reaction with loricrin is also observed in the band at a molecular weight of30 kDa but only in the parakeratinized epithelium in the turkey (Figure 21b, Table 5).

The strong reaction with TGM-1 in the orthokeratinized epithelium in all examined bird species is observed in the bands at 150 kDa (Figure 22a, Table 5). The parakeratinized epithelium polypeptide bands at 50 kDa have a weak reaction to the TGM-1 antiserum in only the goose and duck (Figure 22b, Table 5), while the bands at 80 kDa stain weakly with the TGM-1 antiserum in only the turkey (Figure 22b, Table 5).

## 4. Discussion

The conducted immunohistochemical analysis showed interspecies differences in the distribution of proteins involved in cornification processes, i.e., alpha-keratin, filaggrin, loricrin, and TGM-1, in two studied types of lingual epithelia in the domestic duck, goose, and turkey.

The distribution of alpha-keratin and loricrin in both the ortho- and parakeratinized epithelium, as well as TGM-1 in the orthokeratinized epithelium, is the most homogeneous among the three studied bird species. Only slight interspecies differences in the intensity of color reactions were noted. On the other hand, the pattern of filaggrin distribution in both types of the cornified epithelia of the tongue and the TGM-1 of the parakeratinized epithelium differs between the studied species. The goose is a species with a species-specific distribution pattern of filaggrin in the orthokeratinized epithelium and TGM-1 in the parakeratinized epithelium in comparison to the duck and turkey.

Generally, the present IHC staining reaction with KAPs and TGM-1 in the studied bird species occurs in a granular pattern in the lower epithelial layers of the ortho- and parakeratinized epithelium and in a diffuse pattern in the cornified layer. Similar observations were made in the epithelia of the oral cavity in humans [32]. The only exception in current studies where the diffuse pattern of staining reaction was observed in all epithelial layers was the reaction with filaggrin and loricrin in the parakeratinized epithelium in turkey and with loricrin in the orthokeratinized epithelium in all studied bird species.

IHC studies on the distribution of alpha-keratin in the epithelium of the lingual mucosa in birds have so far been carried out only in chickens [24]. These studies showed that alpha-keratin is present mainly in the basal layer in the parakeratinized epithelium, and weak expression has been demonstrated in the cornified layer. In the orthokeratinized epithelium, alpha-keratin was observed in all epithelial layers; however, in the basal layer, the expression was the weakest. The results of the present analysis show that (i) alpha-keratin is accumulated mainly in the basal and intermediate layers in both epithelia of the tongue in the duck, goose, and turkey; (ii) the cornified layer is characterized by weak expression in both epithelia; and (iii) the intensity of the staining reaction in the cornified layer is higher in the parakeratinized epithelium than in the orthokeratinized epithelium. A comparative analysis of the distribution of alpha-keratin in the cornified epithelia of the tongue in birds indicates the existence of interspecies differences between the chicken and the three studied species of birds. Additionally, the current results confirm the earlier report by Skieresz-Szewczyk et al. [25,26] determining a higher percentage of alpha-keratin in the basal and intermediate layers of the epithelium compared to the cornified layer of both types of cornified epithelia.

Previous biochemical analyses of the alpha-keratin in the epithelium of the lingual mucosa of the avian tongue are also sparse and limited to studies in chickens [24] and parrots [33]. The results in chicken showed that, in the para- and orthokeratinized epithelium, major polypeptides are present at a molecular weight of 47 and 55.5–60 kDa, while in the orthokeratinized epithelium, there are additional polypeptides at 59 and 62 kDa [24]. In turn, in parrots, in the orthokeratinized epithelium, major bands of the alpha-keratin are at polypeptides bands of 15.5 kDa, 56, and 69 kDa and minor bands at 28 kDa [33].

The results of the current biochemical analysis of alpha-keratin indicate that in duck, goose, and turkey, different epitopes of this protein are present in the ortho- and parakeratinized epithelium than in chickens and parrots. The major polypeptide band in both epithelia is present at a molecular weight of 45 kDa, and we revealed the presence of other bands at 57–60 kDa. Additionally, in the parakeratinized epithelium, the major polypeptide band is also present at 47 kDa.

Comparing these results with the previous reports in the apteric and scale epidermis in birds and epidermis in reptiles, we can state similarities in the molecular weight of the alpha-keratin, which, in the lingual epithelium and epidermis, are generally between 40 and 70 kDa [16,34,35].

The cytokeratins are characterized by different mechanical and non-mechanical functions [3,6,36,37]. Alpha-keratin in the apteric and scale epidermis in birds and the scales of reptiles occurs in the lower layers of the epidermis but is absent in the cornified layer or its expression is weak [15,16,34,35,38,39,40]. In that localization, the alpha-keratin is responsible for the mechanical strength of epithelial cells, their adhesiveness, and changes in shape when stretched [39]. In the cornified epithelia of the lingual mucosa in birds, it has been established that alpha-keratin is responsible for forming a cell cytoskeleton, providing an adequate connection between neighboring cells and with the basal membrane [25,26].

When discussing the distribution of alpha-keratin in both types of cornified epithelia, it is worth mentioning the previous studies on the coexistence of alpha-keratin with beta-keratin. Carver and Sawyer [24] showed that beta-keratin is present in single cells of the cornified layer of the parakeratinized epithelium and in the intermediate and cornified layer of the orthokeratinized epithelium. Recent studies by Skieresz-Szewczyk et al. [25,26] using Raman microspectroscopy determined that beta-keratin is synthesized in the lower layers of the ortho- and parakeratinized epithelium of the tongue in goose and is accumulated in the cornified layer of these epithelia, whereby the percentage amount of the beta-keratin in the cornified layer is higher in the orthokeratinized epithelium (70%) than in the parakeratinized epithelium (61%) [25,26]. It should be added that, according to Alibardi [15], beta-keratin is built on a scaffold composed of alpha-keratin, which causes its masking and weaker color reactions with alpha-keratin. The present results confirm these observations.

Molecular analysis of beta-keratin in the tongue epithelium of the chicken showed the presence of an array of polypeptides at 16, 18.5, 19.5, 21, and 24 kDa, which fall within the range of the beta-keratin typical for the scale epidermis in birds and reptiles [16,24,34,35]. Genetic studies indicate that, due to the different genomic locus and gene structure, as well as the amino acid structure, beta-keratin resembles mammalian KAPs and cannot be classified in the keratin family [41]. Thus, beta-keratin as KAPs exhibits limited extensibility, microbiological resistance, and hydrophobicity, and is responsible for the mechanical resistance of the epithelium [39,40,42,43]. The fact that there is more beta-keratin in the cornified layer of the orthokeratinized epithelium than in the parakeratinized epithelium determines that this epithelium is characterized by a higher mechanical resistance [26]. It should be mentioned that the orthokeratinized epithelium of the studied bird species forms the lingual nail and covers the conical papilla of the tongue, so those special structures of the lingual mucosa participate in food intake and require better mechanical protection than the parakeratinized epithelium, which covers the dorsal surface of the tongue in birds, where the transport of food into the esophagus takes place [20,21,22,23].

The other proteins involved in the process of cornification of the epithelium are the so-called keratin-associated proteins, i.e., filaggrin and loricrin [3]. These proteins in mammals are accumulated in the cytoplasm of keratinocytes of the granular layer of the epidermis in the form of keratohyalin granules [3,8,12]. In birds, the apteric and scale epidermis, as well as the para- and orthokeratinized epithelium of the lingual mucosa, do not have keratohyalin granules [15,21,22,23]. However, IHC studies of the epidermis in birds have shown the presence of KAPs, both filaggrin and loricrin, in the keratinocyte cytoplasm of the transitional layer and the lowermost part of the cornified layer [15,16].

The results of the current IHC analysis are the only reports of the distribution of filaggrin and loricrin in the cornified epithelia of the lingual mucosa in birds.

Filaggrin in the orthokeratinized epithelium of the tongue is present only in the intermediate layer. We observed interspecies differences within this layer. Only in goose is filaggrin present in both parts of the intermediate layer, and in duck and turkey, it is only present in the upper part of the intermediate layer. On the other hand, in the parakeratinized epithelium, filaggrin is present both in the intermediate layer and in the cornified layer in all examined bird species. The presence of filaggrin was also noted in the basal layer, but only in duck and goose.

In the literature, data on KAPs in the epithelium of the oral cavity in mammals concern only humans [32]. They showed that filaggrin is present in the upper part of the stratum spinosum, stratum granulosum, and stratum corneum, both in the orthokeratinized epithelium of the palate and the parakeratinized epithelium of the gingiva [32].

Filaggrin is formed during the proteolysis of the high-molecular-weight precursor of profilaggrin, of which molecular weight is over 220 kDa in the human epidermis and more than 300 kDa in rodents [44,45]. The molecular weight of filaggrin in the human epidermis is 35–37 kDa, in mice is 30 kDa, and in rats is 48 kDa [44,46,47]. In the epidermis of birds, bands identified with the anti-filaggrin antibody are in the range of 38 kDa, 48–50 kDa in the apteric epidermis, and 34 kDa and 42–45 kDa in the scale epidermis [16]. Meanwhile, in reptiles, filaggrin bands are located at 55–57 kDa, 66–67 kDa in crocodiles, and 62–64 kDa in snakes [34,35].

Current molecular studies of the lingual epithelia in duck, goose, and turkey showed the presence of filaggrin polypeptide bands mainly at 30 kDa, which is in close range of the mammalian and birds filaggrin [16,44,46,47]. There are also bands over 220 kDa, which are typical of profilaggrin in mammals [44,45]. These studies also reveal the presence of filaggrin-positive bands at 70 kDa, 100 kDa, and 110 kDa, which had not been previously described and could be an epitope characteristic for the ortho- and parakeratinized epithelium of the tongue in birds.

Filaggrin, as a KAPs protein during the process of cornification of the epidermis of mammals, is responsible for joining cytokeratin filaments into thick bundles that fill the keratinocyte cytoplasm of the stratum corneum and may also play a role in cell nucleus apoptosis [9,10].

Filaggrin, present in the intermediate layer of the para- and orthokeratinized epithelium of the lingual mucosa in birds, coexists with the bundles of cytokeratin filaments composed of alpha- and beta-keratin. As previously mentioned, beta-keratin is included in the KAPs [41]. Cytokeratin filaments made of beta-keratin are built upon cytokeratin filaments composed of alpha-keratin and, together with filaggrin, may be responsible for the accumulation and assembly of cytokeratin filaments in the cornified layer. At the same time, as indicated by Dale et al. [48], the presence of filaggrin in the cornified layer is masked, and the current analysis of the cornified epithelia of the tongue in birds has shown that the IHC reactions in the cornified layer are weak, or no color reaction has been demonstrated. Studies on the ultrastructural level of both types of cornified epithelia of the tongue in birds have shown that during the differentiation of keratinocytes in the lower parts of the cornified layer of the orthokeratinized epithelium and in the cornified layer of the parakeratinized epithelium, cell nuclei are still present [49].

Immunohistochemical analyses of the loricrin, performed in the present study, indicate that it occurs in all layers of the epithelium, both ortho- and parakeratinized, in the three studied species of birds. In general, it was found that the intermediate layer of the orthokeratinized epithelium in all investigated bird species was characterized by stronger expression compared to the parakeratinized epithelium, in which weak reactions were noted in the cells of the upper part of the intermediate layer, located just below the cornified layer.

Biochemical analysis of the loricrin in the epidermis of birds showed that the major polypeptide bands for the loricrin are present at 48–50 kDa in the scale and apteric epidermis, at 37–38 kDa and 52–54 kDa in the scale epidermis, and 64–66 kDa in the apteric epidermis [16,50]. These studies also showed the presence of the minor bands at 58–60 kDa both in the apteric and scale epidermis [16,50]. In turn, in the epidermis of reptiles, the molecular weight of the loricrin is mainly in the range of 50–58 kDa [16,33]. There were also minor polypeptides bands observed at 25–27 kDa and 40–42 kDa in lizards, 33 kDa and 48–50 kDa in snakes, and 57–66 kDa and 70–72 kDa in crocodiles [34,35,50]. In mammals in the epidermis, loricrin polypeptides bands are located at 30–35 kDa in humans and sheep and 60 kDa in rodents, rabbits, and cows [51,52].

The conducted WB analyses in duck, goose, and turkey in the lingual epithelia showed that the molecular weight of the major bands of loricrin range between 33 and 35 kDa and are also at 30 kDa. Thus, in the cornified epithelia of the avian tongue, the molecular weight of loricrin is in the same range as the mammalian loricrin [51,52]. In three examined bird species, we also observed a minor band at 60 kDa in the orthokeratinized epithelium, which is characteristic of apteric and scale skin in birds, and the epidermis in crocodiles, rodents, rabbits, and cows [16,35,50,52].

Comparing the intensity expression of the filaggrin and loricrin in the lingual epithelia in birds, we can conclude that they are stronger in the orthokeratinized epithelium than in the parakeratinized epithelium. A similar observation has been revealed in the epithelia of the human oral cavity [32]. Loricrin compared to filaggrin is localized in all epithelial layers, both in the ortho- and parakeratinized epithelium, and additionally, their expression is stronger than that of filaggrin. These features indicate that the major keratin-associated protein is loricrin. This fact may constitute a feature differentiating the epithelium of the beak cavity from the epidermis in birds, where the expression of filaggrin and loricrin is the same [16].

In addition, microscopic observations of Masson–Goldner topographic staining in the parakeratinized epithelium and immunohistochemical staining with alpha-keratin, filaggrin, and loricrin performed in this epithelium showed interesting relationships, indicating the possibility of diagnostic features of topographic staining and its potential importance before performing further immunohistochemical analyses. The cell cytoplasm of the superficial cells of the cornified layer, which stains pink during Masson–Goldner staining, showed stronger reactions with these proteins than the lower cells, with the cytoplasm stained red. An exception in this regard is the parakeratinized epithelium of turkey, which shows a generally weaker color reaction after staining with loricrin.

Besides KAPs, the enzyme TGM-1 is synthesized in the cytoplasm of keratinocytes, which, in the epidermis of mammals, occurs mainly in the spinous and granular layer [53,54,55]. Weak color reactions were also seen in the basal layer [53,55]. Immunoreactivity was detected in the plasma membrane in the granular and outer spinous layers or in the cell cytoplasm in the upper spinous layer [54]. Studies on the avian epidermis indicate that TGM-1 is present in the transitional layer and the lowermost part of the corneous layer only in the apteric skin [15,16,50]. In the scale epidermis in birds and in scales in reptiles, TGM-1 does not occur [16,50].

Current IHC studies with TGM-1 indicate their presence in both types of cornified epithelia of the tongue in duck, goose, and turkey. Immunoreactivity was observed only in the cell cytoplasm, and we do not observe the expression in the plasma membrane as it is in humans [54]. TGM-1 in the orthokeratinized epithelium is present in all layers in the three studied species of birds, and in the parakeratinized epithelium in all layers of the epithelium, but only in duck and turkey. In the domestic goose, TGM-1 occurs only in 2–3 layers of superficial cells in the cornified layer. TGM1, similar to loricrin and filaggrin, is characterized by weaker expression in the parakeratinized epithelium than in the orthokeratinized epithelium.

At this point, attention should be paid to the special microstructure of the parakeratinized epithelium in the domestic turkey, which is characterized by the presence of long connective tissue papillae reaching the beginning of the cornified layer. Their presence determines the differentiation and shifting of keratinocytes towards the epithelial surface, which disturbs the typical horizontal arrangement of keratinocytes into layers, and between the connective tissue papillae, cells of the basal and lower part of the intermediate layer are present. Such a structure of the parakeratinized epithelium in turkey causes the color reactions with filaggrin, loricrin, and TGM-1 described below; in the area of the lower part of the intermediate layer, colored bands appear arranged between the connective tissue papillae.

TGM-1 is a 92 kDa protein in the human epidermis and is present in the range of 44–50 kDa, 55–58 kDa, 60–64 kDa, and 67–70 kDa in rats [50,55,56], whereas, in the epidermis of reptiles, polypeptide bands of TGM-1 are located at 40–55 kDa and 57–60 kDa [50]. In the hard epidermis of the shell of the turtle, a minor band at 130 kDa was also detected [50]. The apteric and scale epidermis in birds is characterized by TGM-1 polypeptide major bands in the range of 50–58 kDa and 70–72 kDa and minor bands at 42–48 kDa [16,50].

The obtained results of the molecular analysis in the ortho- and parakeratinized epithelium of the tongue in birds revealed the presence of a 150 kDa band in the orthokeratinized epithelium, which might be a new epitope in close range of the epidermis TGM-1 in turtles [50]. Meanwhile, in the parakeratinized epithelium, we detected TGM-1 at 50 kDa, which is an epitope similar to the TGM-1 in the apteric and scale epidermis in birds. In the turkey, we observed unique polypeptide bands at 80 kDa, which were not observed previously.

Loricrin as KAPs in mammals, together with involucrin, SPR, cystatin, and elafin, desmoplakin, envoplakin, periplakin, and the S100 protein, forms a protein complex called the cornified cell envelope [10,11]. This complex is anchored to the keratinocyte cell membrane byTGM-1. TGM-1 belongs to the family of enzymes that catalyze the formation of isopeptides bonds, and its activity is regulated by calcium, the concentration of which increases from the basal layer towards the granular layer [57,58]. The presence of loricrin and TGM-1 in the ortho- and parakeratinized epithelium of the tongue in birds indicates the possibility of the formation of a protein complex of the cornified cell envelope in the cytoplasm of keratinocytes.

## 5. Conclusions

To sum up, alpha-keratin, filaggrin, loricrin, and TGM-1, i.e., the proteins responsible for the process of cornification of the multi-layered epithelia, are expressed in the cell cytoplasm of the ortho- and parakeratinized epithelium of the lingual mucosa in birds. Our research allowed us to determine the pattern of occurrence of alpha-keratin and study KAPs and TGM-1 in both types of epithelia. The orthokeratinized epithelium is characterized by stronger expression with filaggrin, loricrin, and TGM-1 and weaker expression with alpha-keratin compared to the parakeratinized epithelium. These facts prove that both types of lingual epithelia in birds undergo cornification, and the orthokeratinized epithelium is characterized by a more advanced process of cornification compared to the parakeratinized epithelium. At the same time, it may determine the formation of an efficient protective mechanical barrier of the orthokeratinized epithelium in comparison to the parakeratinized epithelium, resulting from the location of these epithelia on the lingual mucosa of the avian tongue and in the context of food intake and transport to the esophagus. Biochemical analyses of the studied proteins indicate that epitopes of alpha-keratin and TGM-1 present in the cornified epithelia of the studied birds’ tongues are typical of the epidermis of birds and reptiles. The exception is TGM-1 in turkeys, showing polypeptide bands unique to other vertebrates’ molecular weight. Loricrin epitopes defined in the ortho- and parakeratinized epithelium of the tongue in studied birds are characteristic of the epidermis of mammals, birds, and reptiles. In the case of filaggrin, the determined epitopes are within the mammalian epidermal filaggrin, and the remaining epitopes are unique to birds. The above conclusions, indicating the presence of proteins characteristic of epidermal cornification in the ortho- and parakeratinized epithelium of birds’ tongues, may prove that these epithelia, similarly to the epidermis, are of ectodermal origin. At the same time, our research indicates that, unlike the epidermis, the main KAPs involved in the cornification process of the epithelium of birds’ tongues is loricrin.

## Figures and Tables

**Figure 1 cells-11-01899-f001:**
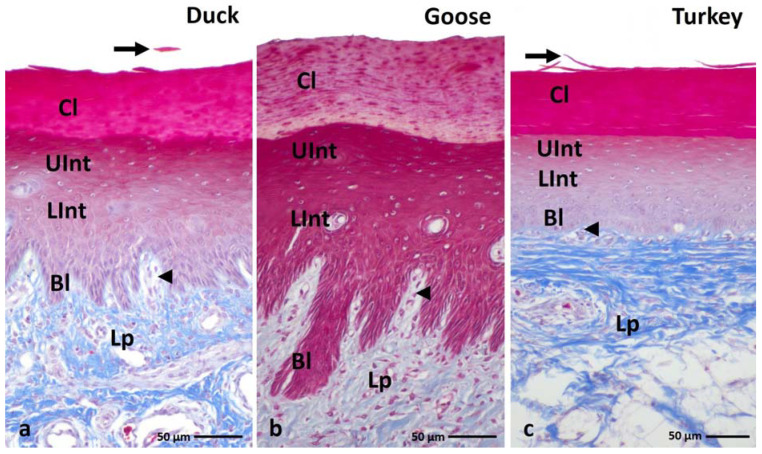
(**a**–**c**) Cross-section of the **orthokeratinized epithelium.** Black arrows show exfoliated superficial cells of the cornified layer. Arrowheads point to the connective tissue papillae. Bl—basal layer, Lp—lamina propria, LInt—the lower part of the intermediate layer, UInt—upper part of the intermediate layer, Cl—cornified layer. Masson–Goldner staining.

**Figure 2 cells-11-01899-f002:**
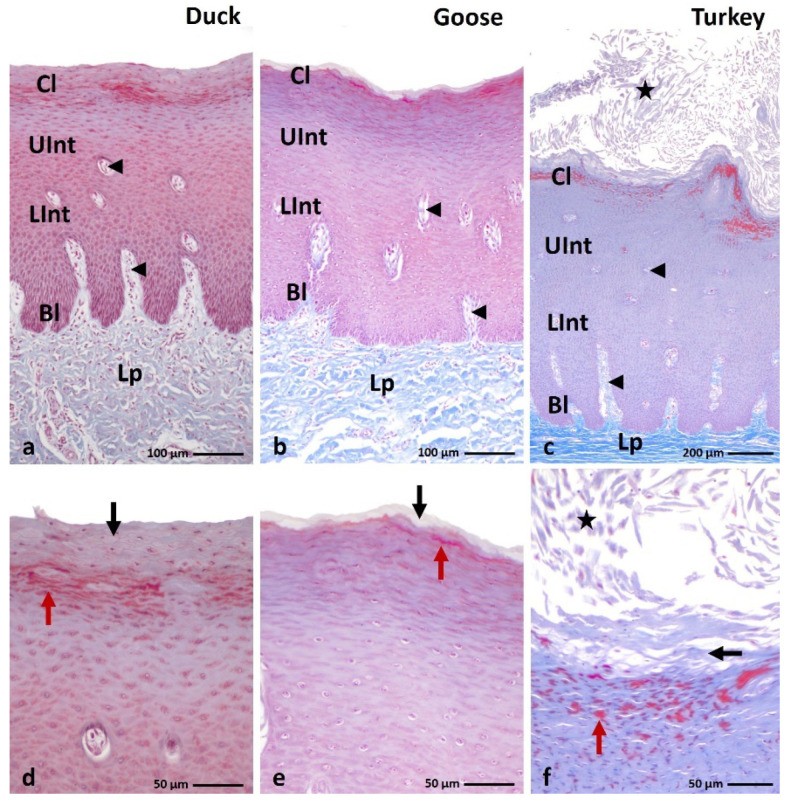
(**a**–**c**) Cross-section of **the parakeratinized epithelium.** Arrowheads point to the connective tissue papillae. Asterisks show massively exfoliated superficial cells of the cornified layer. (**d**–**f**) Higher magnification of the cornified layer of the parakeratinized epithelium. Black arrows show superficial cells of the cornified layer, which cytoplasm stained with pink or light pink color. Red arrows point to cells with red-colored cell cytoplasm of the cornified layer. Bl—basal layer, Lp—lamina propria, LInt—the lower part of the intermediate layer, UInt—upper part of the intermediate layer, Cl—cornified layer. Masson–Goldner staining.

**Figure 3 cells-11-01899-f003:**
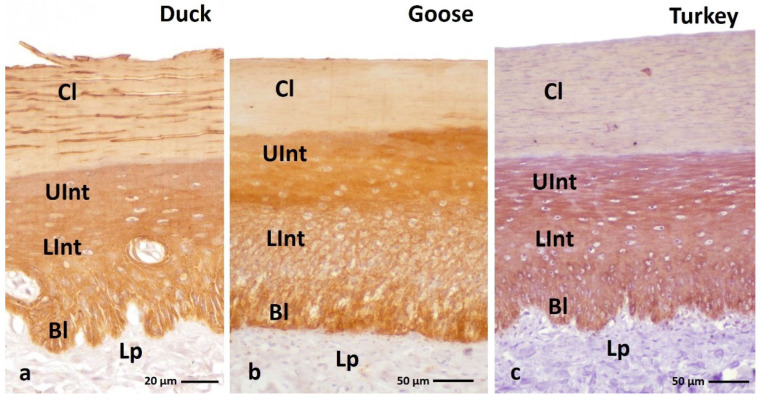
(**a**–**c**) IHC staining of the **alpha-keratin** in the **orthokeratinized epithelium.** Bl—basal layer, Lp—lamina propria, Lint—the lower part of the intermediate layer, UInt—upper part of the intermediate layer, Cl—cornified layer. Counterstaining with Mayer’s hematoxylin.

**Figure 4 cells-11-01899-f004:**
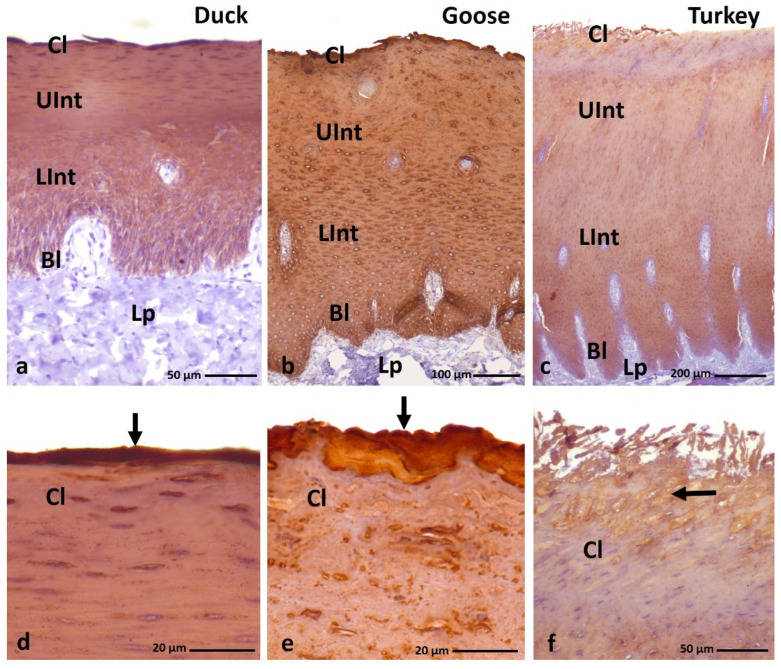
(**a**–**c**) IHC staining of the **alpha-keratin** in the **parakeratinized epithelium**. (**d**–**f**) Higher magnification of the cornified layer. Arrows point to the superficial cells of the cornified layer with a strong staining reaction. Bl—basal layer, Lp—lamina propria, LInt—lower part of the intermediate layer, UInt—upper part of the intermediate layer, Cl—cornified layer. Counterstaining with Mayer’s hematoxylin.

**Figure 5 cells-11-01899-f005:**
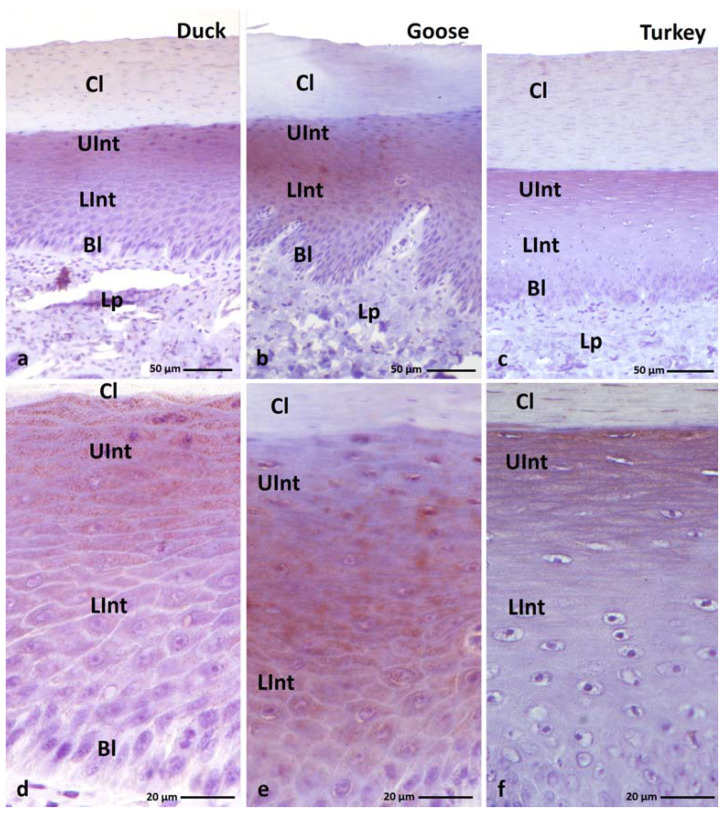
(**a**–**c**) IHC staining of the **filaggrin** in the **orthokeratinized epithelium.** (**d**–**f**) Higher magnification of basal, intermediate, and cornified layers of the orthokeratinized epithelium. Bl—basal layer. Lp—lamina propria, LInt—the lower part of the intermediate layer, UInt—upper part of the intermediate layer, Cl—cornified layer. Counterstaining with Mayer’s hematoxylin.

**Figure 6 cells-11-01899-f006:**
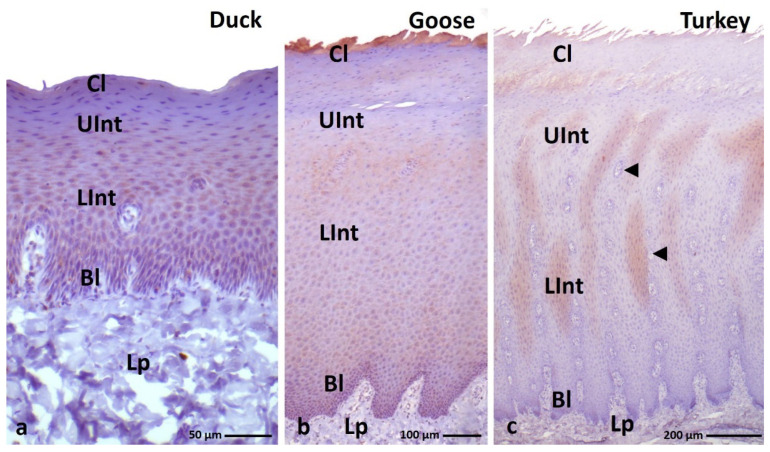
(**a**–**c**) IHC staining of the **filaggrin** in the **parakeratinized epithelium.** Arrowheads point to the connective tissue papillae. Bl—basal layer, Lp—lamina propria, LInt—the lower part of the intermediate layer, UInt—upper part of the intermediate layer, Cl—cornified layer. Counterstaining with Mayer’s hematoxylin.

**Figure 7 cells-11-01899-f007:**
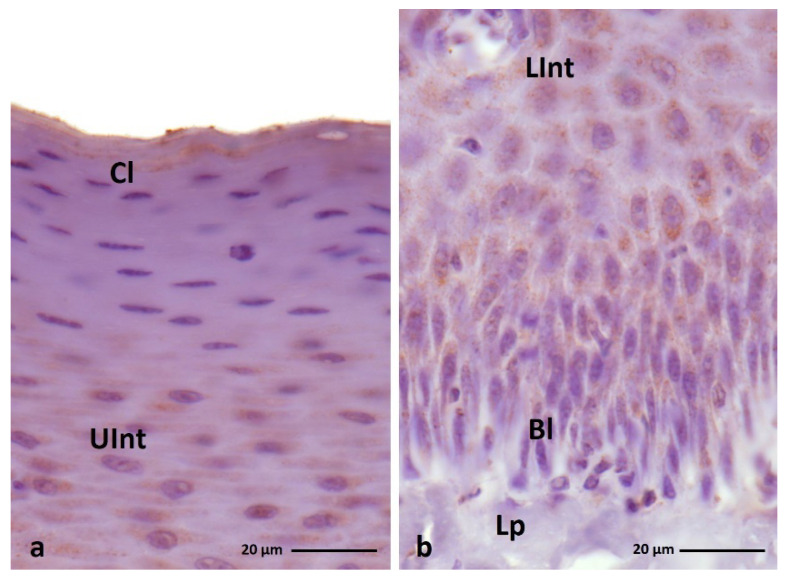
IHC staining of the **filaggrin** in the **parakeratinized epithelium** in **duck**. Higher magnification of (**a**) the upper part of the intermediate layer and cornified layer and (**b**) the basal layer and lower part of the intermediate layer. Bl—basal layer, Lp—lamina propria, LInt—lower part of the intermediate layer, UInt—upper part of the intermediate layer, Cl—cornified layer. Counterstaining with Mayer’s hematoxylin.

**Figure 8 cells-11-01899-f008:**
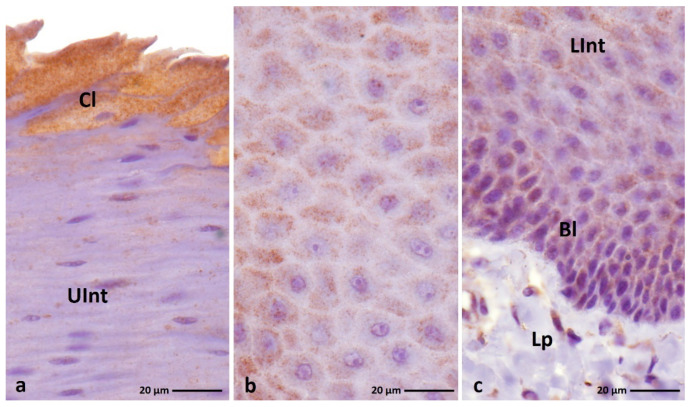
IHC staining of the **filaggrin** in the **parakeratinized epithelium** in **goose.** Higher magnification of (**a**) the upper part of the intermediate layer and cornified layer, (**b**) the lower part of the intermediate layer, and (**c**) the lower part of the intermediate layer and basal layer. Bl—basal layer, Lp—lamina propria, LInt—lower part of the intermediate layer, UInt—upper part of the intermediate layer, Cl—cornified layer. Counterstaining with Mayer’s hematoxylin.

**Figure 9 cells-11-01899-f009:**
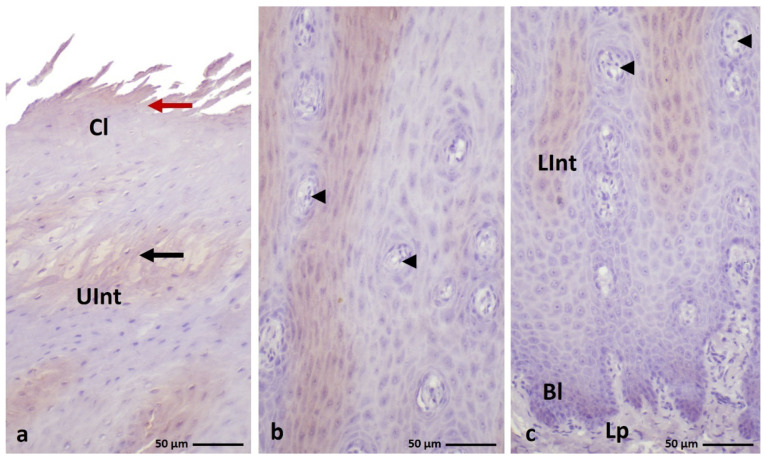
IHC staining of the **filaggrin** in the **parakeratinized epithelium** in the **turkey.** Higher magnification of (**a**) the upper part of the intermediate layer and cornified layer, (**b**) the lower part of the intermediate layer, and (**c**) the lower part of the intermediate layer and basal layer. The black arrow points to the 2–3 cell layers of the upper part of the intermediate layer with a weak staining reaction. The red arrow shows the superficial cells of the cornified layer with a weak staining reaction. Arrowheads point to the connective tissue papillae. Bl—basal layer, Lp—lamina propria, LInt—lower part of the intermediate layer, UInt—upper part of the intermediate layer, Cl—cornified layer. Counterstaining with Mayer’s hematoxylin.

**Figure 10 cells-11-01899-f010:**
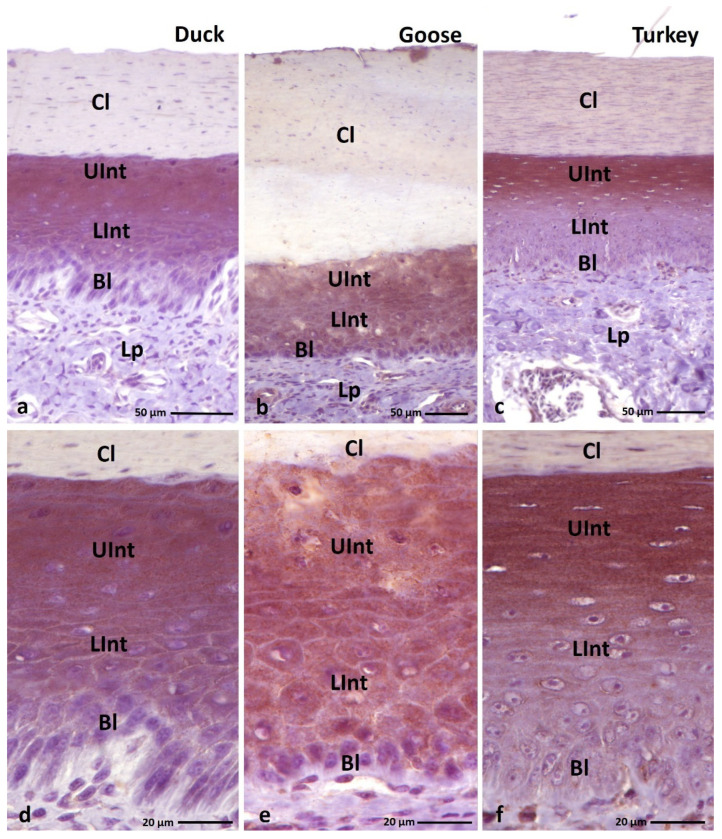
(**a**–**c**) IHC staining of the **loricrin** in the **orthokeratinized epithelium.** (**d–f**) Higher magnification of basal, intermediate, and cornified layers of the orthokeratinized epithelium. Bl—basal layer, Lp—lamina propria, LInt—lower part of the intermediate layer, UInt—upper part of the intermediate layer, Cl—cornified layer. Counterstaining with Mayer’s hematoxylin.

**Figure 11 cells-11-01899-f011:**
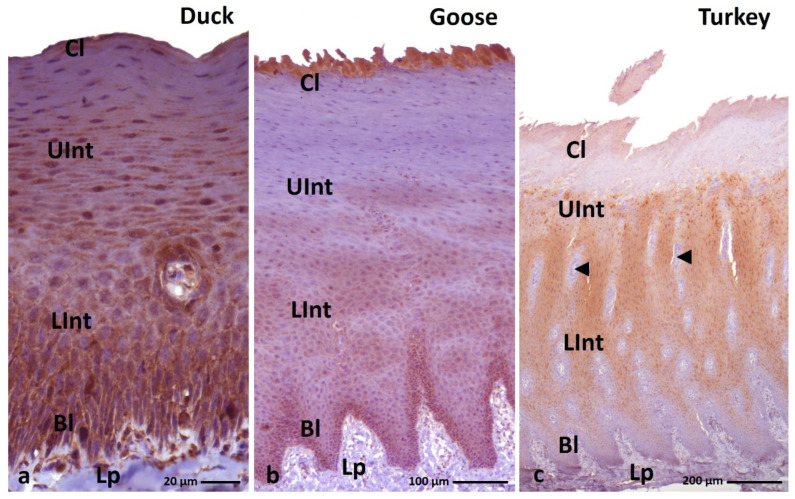
(**a**–**c**) IHC staining of the **loricrin** in the **parakeratinized epithelium.** Arrowheads point to the connective tissue papillae. Bl—basal layer, Lp—lamina propria, LInt—the lower part of the intermediate layer, UInt—upper part of the intermediate layer, Cl—cornified layer. Counterstaining with Mayer’s hematoxylin.

**Figure 12 cells-11-01899-f012:**
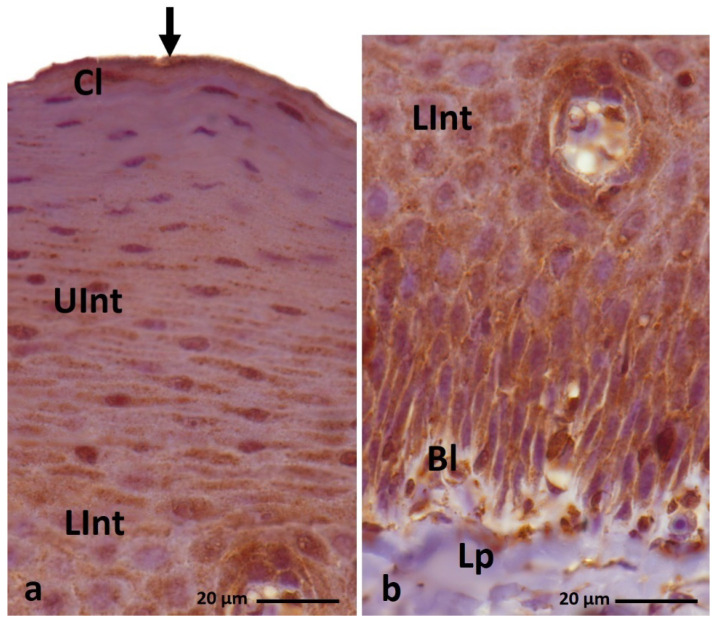
IHC staining of the **loricrin** in the **parakeratinized epithelium** in the **duck.** Higher magnification of (**a**) the intermediate layer and cornified layer and (**b**) the lower part of the intermediate layer and basal layer. Arrow shows the superficial cell with a strong staining reaction. Bl—basal layer, Lp—lamina propria, LInt—lower part of the intermediate layer, UInt—upper part of the intermediate layer, Cl—cornified layer. Counterstaining with Mayer’s hematoxylin.

**Figure 13 cells-11-01899-f013:**
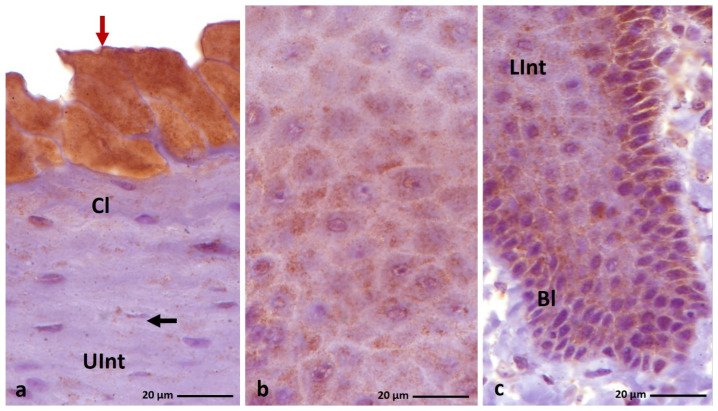
IHC staining of the **loricrin** in the **parakeratinized epithelium** in the **goose.** Higher magnification of (**a**) the upper part of the intermediate layer and cornified layer, (**b**) the lower part of the intermediate layer, and (**c**) the basal layer and lower part of the intermediate layer. Red arrow points to the superficial cell with a strong staining reaction. Black arrow points to the cell of the upper part of the intermediate layer with a weak staining reaction. Bl—basal layer, Lp—lamina propria, LInt—lower part of the intermediate layer, UInt—upper part of the intermediate layer, Cl—cornified layer. Counterstaining with Mayer’s hematoxylin.

**Figure 14 cells-11-01899-f014:**
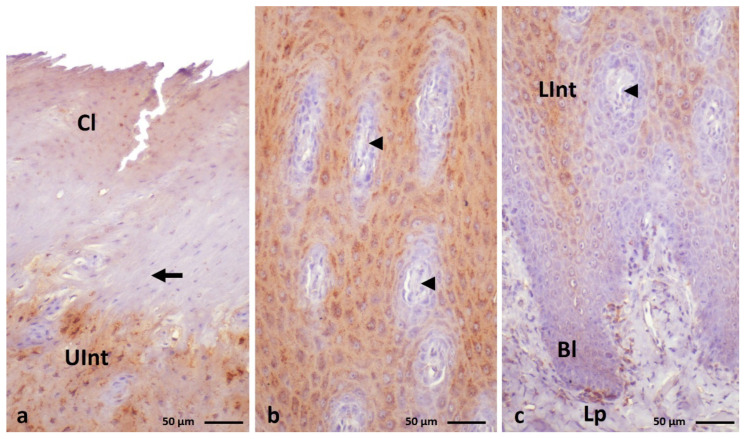
IHC staining of the **loricrin** in the **parakeratinized epithelium** in the **turkey.** Higher magnification of (**a**) the upper part of the intermediate layer and cornified layer, (**b**) the lower part of the intermediate layer, and (**c**) the basal layer and lower part of the intermediate layer. The black arrow points to the cells of the intermediate layer with weak staining reaction. Arrowheads point to the connective tissue papillae. Bl—basal layer, Lp—lamina propria, LInt—the lower part of the intermediate layer, UInt—upper part of the intermediate layer, Cl—cornified layer. Counterstaining with Mayer’s hematoxylin.

**Figure 15 cells-11-01899-f015:**
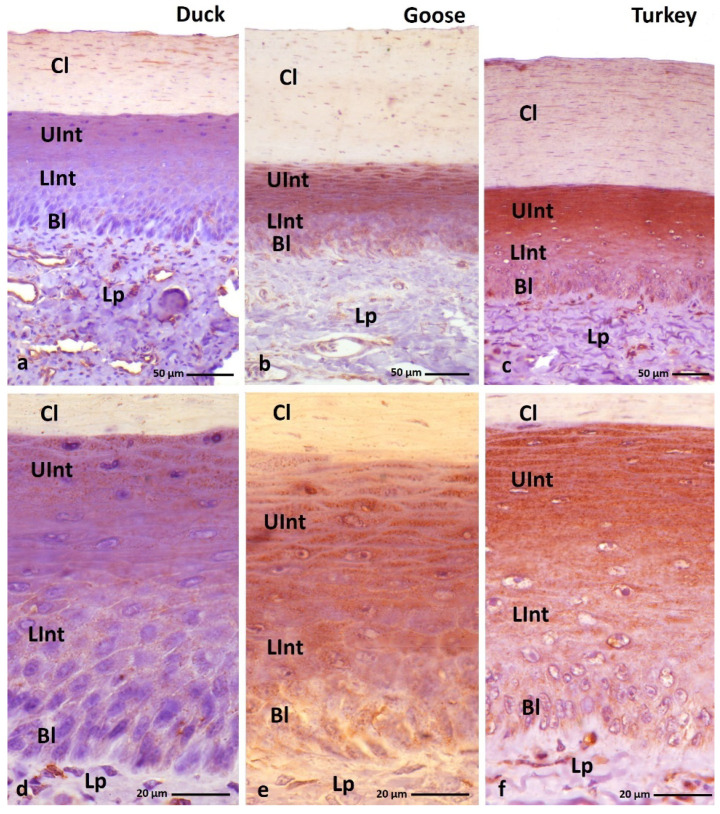
(**a**–**c**) IHC staining of the TGM-1 in the **orthokeratinized epithelium.** (**d**–**f**) Higher magnification of basal, intermediate, and cornified layers of the orthokeratinized epithelium. Bl—basal layer, Lp—lamina propria, LInt—the lower part of the intermediate layer, UInt—upper part of the intermediate layer, Cl—cornified layer. Counterstaining with Mayer’s hematoxylin.

**Figure 16 cells-11-01899-f016:**
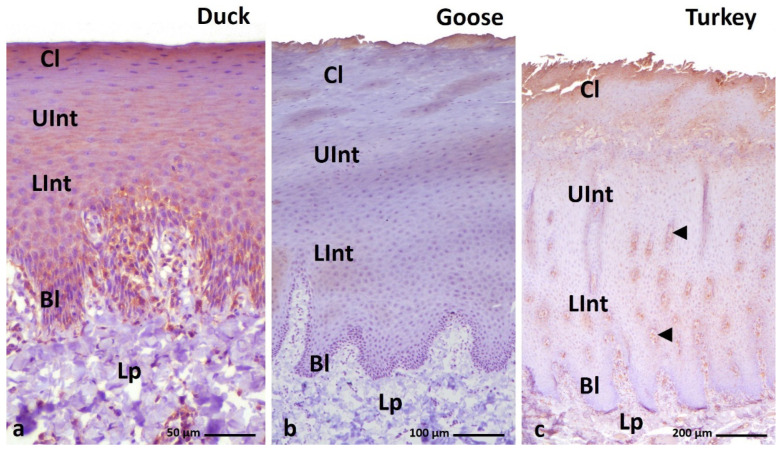
(**a**–**c**) IHC staining of the **TGM-1** in the **parakeratinized epithelium.** Arrowheads point to the connective tissue papillae. Bl—basal layer, Lp—lamina propria, LInt—the lower part of the intermediate layer, UInt—upper part of the intermediate layer, Cl—cornified layer. Counterstaining with Mayer’s hematoxylin.

**Figure 17 cells-11-01899-f017:**
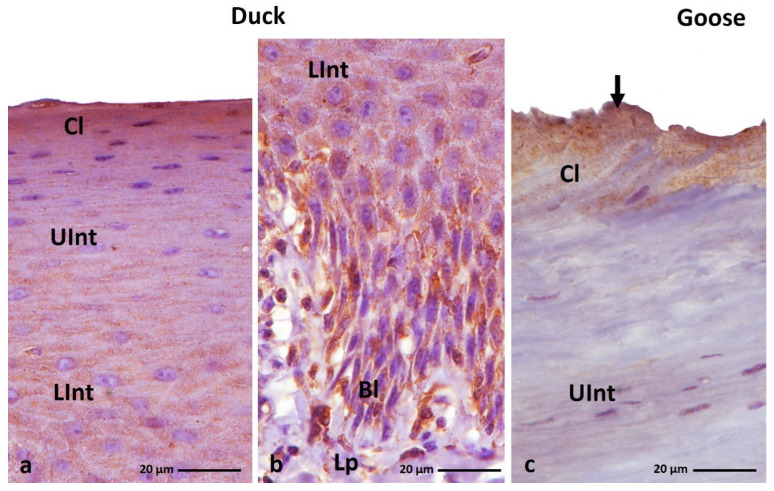
IHC staining of the **TGM-1** in the **parakeratinized epithelium.** Higher magnification of (**a**) the upper part of the intermediate layer and cornified layer in the duck, (**b**) the basal layer and lower part of the intermediate layer in the duck, and (**c**) the upper part of the intermediate layer and cornified layer in the goose. The black arrow points to the superficial cells of the cornified layer with a medium staining reaction. Bl—basal layer, Lp—lamina propria, LInt—lower part of the intermediate layer, UInt—upper part of the intermediate layer, Cl—cornified layer. Counterstaining with Mayer’s hematoxylin.

**Figure 18 cells-11-01899-f018:**
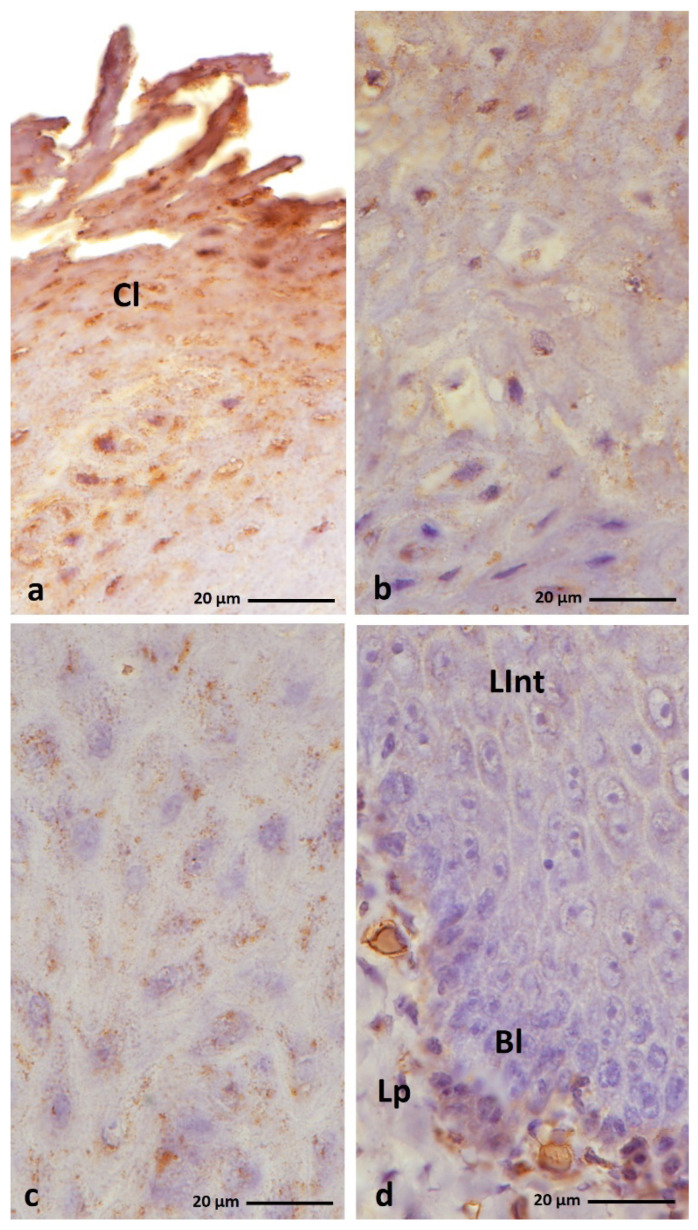
IHC staining of the **TGM-1** in the **parakeratinized epithelium** in the turkey. Higher magnification of (**a**) the cornified layer, (**b**) the upper part of the intermediate layer, (**c**) the lower part of the intermediate layer, and (**d**) the basal layer and lower part of the intermediate layer. Bl—basal layer, Lp—lamina propria, LInt—the lower part of the intermediate layer, UInt—upper part of the intermediate layer, Cl—cornified layer. Counterstaining with Mayer’s hematoxylin.

**Figure 19 cells-11-01899-f019:**
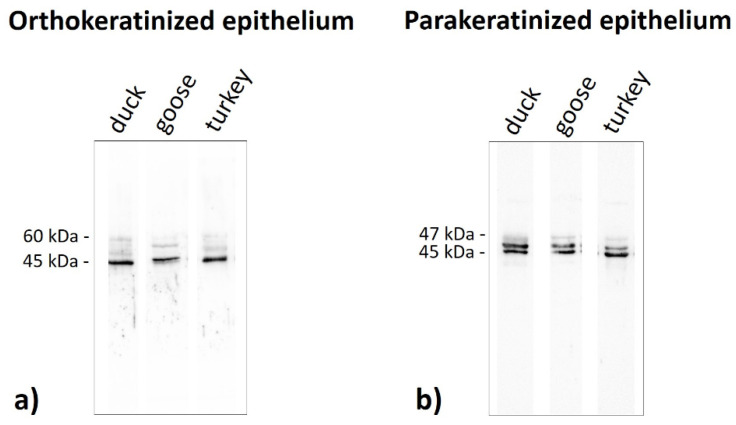
Immunoblots of the (**a**) orthokeratinized and (**b**) parakeratinized epithelium with the **alpha-keratin** antiserum in the duck, goose, and turkey.

**Figure 20 cells-11-01899-f020:**
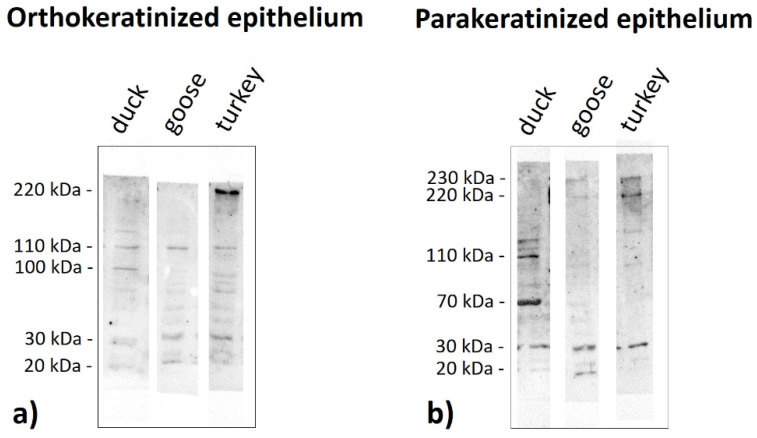
Immunoblots of the (**a**) orthokeratinized and (**b**) parakeratinized epithelium with the **filaggrin** antiserum in the duck, goose, and turkey.

**Figure 21 cells-11-01899-f021:**
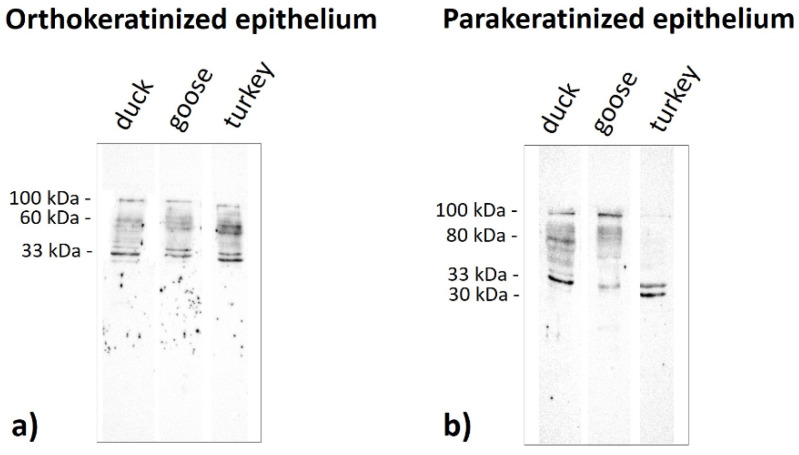
Immunoblots of the (**a**) orthokeratinized and (**b**) parakeratinized epithelium with the **loricrin** antiserum in the duck, goose, and turkey.

**Figure 22 cells-11-01899-f022:**
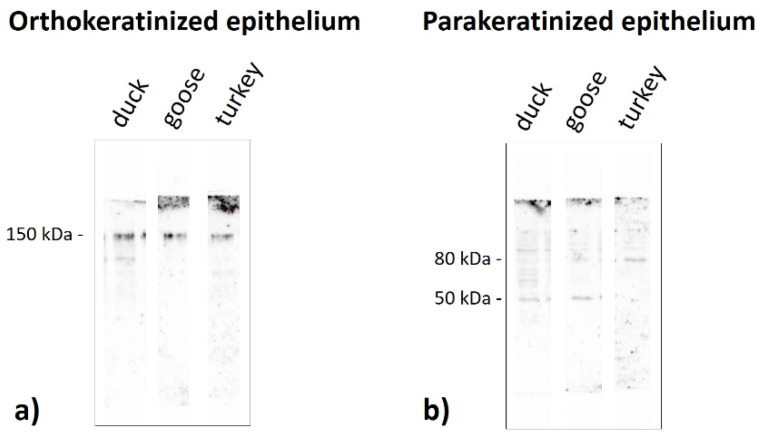
Immunoblots of the (**a**) orthokeratinized and (**b**) parakeratinized epithelium with the TGM-1 in the duck, goose, and turkey.

**Table 1 cells-11-01899-t001:** Spatial distribution of the alpha-keratin in the ortho- and parakeratinized epithelium in duck, goose, and turkey. +++ strong reaction, ++ medium reaction, + weak reaction.

**Orthokeratinized Epithelium**				
**Species**	**Basal Layer**	**Intermediate Layer**		**Cornified Layer**
		**Lower Part**	**Upper Part**	
**Duck**	+++	+++	+++	+
**Goose**	+++	+++	+++	+
**Turkey**	+++	+++	+++	+
**Parakeratinized Epithelium**				
**Species**	**Basal Layer**	**Intermediate Layer**		**Cornified Layer**
		**Lower Part**	**Upper Part**	
**Duck**	+++	+++	+++	++/+++
**Goose**	+++	+++	+++	++/+++
**Turkey**	++	+++	+++	++/+++

**Table 2 cells-11-01899-t002:** Spatial distribution of the filaggrin in the ortho- and parakeratinized epithelium in domestic duck, goose, and turkey. ++ medium reaction, + weak reaction, − lack of IHC reaction.

**Orthokeratinized Epithelium**				
**Species**	**Basal Layer**	**Intermediate Layer**		**Cornified Layer**
		**Lower Part**	**Upper Part**	
**Duck**	−	−	++	−
**Goose**	−	++	−/+/++	−
**Turkey**	−	−	+	−
**Parakeratinized Epithelium**				
**Species**	**Basal Layer**	**Intermediate Layer**		**Cornified Layer**
		**Lower Part**	**Upper Part**	
**Duck**	+	+	+	−/+
**Goose**	+	+	+	+/++
**Turkey**	−	+	−/+	−/+

**Table 3 cells-11-01899-t003:** Spatial distribution of the loricrin in the ortho- and parakeratinized epithelium in the duck, goose, and turkey. +++ strong reaction, ++ medium reaction, + weak reaction.

**Orthokeratinized Epithelium**				
**Species**	**Basal Layer**	**Intermediate Layer**		**Cornified Layer**
		**Lower Part**	**Upper Part**	
**Duck**	+	++	+++	+
**Goose**	+	+++	+++	+
**Turkey**	+	+	+++	+
**Parakeratinized Epithelium**				
**Species**	**Basal Layer**	**Intermediate Layer**		**Cornified Layer**
		**Lower Part**	**Upper Part**	
**Duck**	++	++	++	+/++
**Goose**	++	++	+/++	+/++
**Turkey**	+	++	+/++	+

**Table 4 cells-11-01899-t004:** Spatial distribution of the TGM-1 in the ortho- and parakeratinized epithelium in the duck, goose, and turkey. ++ medium reaction, + weak reaction, − lack of IHC reaction.

**Orthokeratinized Epithelium**				
**Species**	**Basal Layer**	**Intermediate Layer**		**Cornified Layer**
		**Lower Part**	**Upper Part**	
**Duck**	+	+	++	+
**Goose**	+	+	++	+
**Turkey**	+	+	++	+
**Parakeratinized Epithelium**				
**Species**	**Basal Layer**	**Intermediate Layer**		**Cornified Layer**
		**Lower Part**	**Upper Part**	
**Duck**	+	+	+	++
**Goose**	−	−	−	−/++
**Turkey**	−	+	+	+

**Table 5 cells-11-01899-t005:** Molecular weights of the alfa-keratin, filaggrin, loricrin, and TGM-1 in the ortho- and parakeratinized epithelium in the duck, goose, and turkey. +++ strong reaction, ++ medium reaction, + weak reaction.

	**Orthokeratinized Epithelium**			**Parakeratinized Epithelium**		
	**Duck**	**Goose**	**Turkey**	**Duck**	**Goose**	**Turkey**
**Alfa-keratin**	45 kDa (+++) 57–60 kDa (+)			45–47 kDa (+++) 57–60 kDa (++)		
**Filaggrin**	30 and 110 kDa (+++)			30 kDa (+++)		
	100 kDa (+++)		230 kDa (+++)	70 and 110 kDa (+++)		
				20 kDa (+)		
	20 kDa (+)				220 and 230 kDa (+)	
**Loricrin**	33–35 kDa (+++) 60 and 100 kDa (++)			33–35 kDa (+++) 100 kDa (+)		
				80 kDa (+)		30 kDa (+++)
**TGM-1**	150 kDa (+++)			50 kDa (+)		80 kDa (+)

## Data Availability

The data presented in this study are available on request from the corresponding author. The data are not publicly available due to privacy or ethical restrictions.

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
