# Peer review of "Alpha-Keratin, Keratin-Associated Proteins and Transglutaminase 1 Are Present in the Ortho- and Parakeratinized Epithelium of the Avian Tongue"

_cells, 2022, doi:10.3390/cells11121899_

Round 1

Reviewer 1 Report

The manuscript cells-1708227 entitled "Alpha-keratin, keratin-associated proteins and transglutaminase 1 are present in the ortho- and parakeratinized epithelium of the avian tongue, IHC and WB analysis" addressed an interesting argument. The manuscript has been well written; however, a number of concerns emerge that need to be covered, in order to improve the readability of the paper and its scientific soundness. The main criticism is the absence of the statistical analysis section, please explain the motivation for the absence or add it. Moreover, a clear definition of the hypothesis and the novelty of the study have to be introduced in the manuscript. 

Point to point revision has been presented as follows:

Title proposal: I suggest removing the part related to the methodology "Alpha-Keratin, Keratin-Associated Proteins, and Transglutaminase 1 are Present in the Ortho- and Parakeratinized Epithelium of the Avian Tongue"or use a more generic title. 

Introduction section: Please reduce this section, it contains numerous information, not all necessary to explain the rationale of the manuscript. 

L157-159: Please rewrite this sentence, it is not clear the explanation of the samples collection. 

The statistical analysis is missing. Please add it. 

Results section: The table and figures presented are too much and it is not easy to maintain the attention of the reader, in my opinion, the authors have to choose the main representative and create a supplementary file including all the figures and table less important for the data discussion. 

L258: When the authors introduce the tables are invited to describe the results in this section or re-organize the subtitle for the results section. 

Please uniform the term "Table" in the manuscript 

L265-266: How did the authors define strong, medium, and weak reactions? Please add a description.

L501: Please something is missing at the end of the sentence. 

Figure 19: Please use an image with the three merged repetitions in order to have an image more representative of the experiment. If the authors prefer to leave that Figure please delete "the nitrocellulose membranes were cut because each protein for species was repeated three times,". 

Figures 20, 21, and 22: Please use images with optimal quality. Maybe it could be better to choose an inverted image. 

Discussion section: As in the introduction section comment, it is too long, please avoid the references to the introduction section (L797) and discuss more in-depth the results obtained. 

Reviewer 2 Report

Comments on the manuscript "Alpha Keratin, keratin-associated proteins and transglutaminase 1 are present in the ortho- and parakeratinized epithelium of the avian tongue, IHC and WB analysis" by Skieresz-Szewczyk and co-authors.

This paper is a comparative study using the tongue as model system to analyse the protein synthesis during keratinocyte differentiation. The location of several proteins (alpha-keratin, filaggrin, and loricrin) and an enzyme (TGM-1) involved in cornification in the lingual epithelia of three different bird species is described. It is rare to come across such a scholarly work. The authors know the subject well and have a great comparative view on the matter. The only down side is that they have little new to tell us. However, I am so impressed with the quality of the work that I am not going to be hard on them for that reason.

General comments:

It is mandatory to reduce the length of the “Discussion” section. Seems to be very repetitive. Please, include the most striking differences with results obtained in chicken and, finally, compare the results obtained with those described in mammals and reptiles.

Specific comments:

I suggest a more detailed protocol for immunohistochemistry (including DAB reaction).

I suggest to include in the figure legends the counterstaining method used with immunohistochemistry.

Please, include the name of the bird species in the upper right corner of the panels.

Figure3: It is not clear a “diffuse staining reaction”. Strong immunoreactivity is found in the Cl in 3A, and the immunoreactivity in the Bl and in the Llnt is not similar to that found in the Ulnt in 3A and 3B. Immunoreactivity seems to be strong in the Bl. I suggest to describe it more accurately.

Bands of immunoreactivity are seen in Fig. 6C: could it be an artifact?. Again in Fig. 9.

In the “Results” section and Figure legends appear “transglutaminase-1” many times. Please, be consistent with the abbreviation (TGM-1).

Reviewer 3 Report

The article titled “Alpha-keratin, keratin-associated proteins and transglutaminase 1 are present in the ortho- and parakeratinized epithelium of the avian tongue, IHC and WB analysis” deals with an interesting topic and provides a very detailed description of the distribution of some molecules at the level of the para- and orthokeratinized epithelium of the tongue in three avian species.

High-quality histological images are provided.

The introduction, results and discussion are covered in an in-depth and detailed manner.

Some considerations that should be addressed are listed below.

The description of species is not uniform along with the text and figure captions. "Domestic duck, goose and turkey", and "duck, goose and domestic turkey" and "duck and domestic turkey". Please, make it uniform.

Line 55. The citation numbers should be in ascending order.

Line 73. The acronym “IHC” first appears in the text. Please, also write the full name.

Line 153: “clone AE1/AE3”. Along the text it is also written “AE1/3” and “AE1/ 3”. Please, write the abbreviations evenly.

Material and methods: The IHC procedure should be briefly described. Write which chromogen was used for IHC and which colour to contrast the sections. Has the negative control been done? How?

The abbreviation of transglutaminase 1 is differently written along with the text: Anti-TGM1, Anti-TGM-1, TGM1, TGM-1. Please, uniform it. Also, write the acronym next to the full name the first time it appears.

Line 184: Looking at Figure 2c, the parakeratinized epithelium of the domestic turkey seems much taller than 114.3 micrometers. Please check if the measurement indicated in the text in line 184 is correct.

Line 221: Does "the lower part" refer to the intermediate layer? Please, write it.

Line 218-233: The morphological description is accurate but cannot be referred to figures 1 and 2 as the magnification is low. It is necessary to add some photos at least for the more peculiar characteristics (for example the differences observed in the common turkey).

Line 360: “Arrowheads point to the connective tissue papillae” This sentence can be written only once in the caption.

Table 5. Reading and understanding the table is difficult. It is not clear to which species the values refer. For example, the values of alfa-keratin and loricrin seem to refer only to the goose. Filaggrin is very confusing.

Line 501: delete “and”.

Round 2

Reviewer 1 Report

The authors addressed all the reviewer suggestions, and the revised version of the manuscript has been ameliorated. Just a minor check is on the references that seem to need to be edited for the font size.